# Not All Structure Is Learned: Disentangling Inherited and Learned Representations in Recurrent Networks

**Mark Alence**                                                    *mark.alence@gmail.com*
*Independent*

**Reviewed on OpenReview:** *https://openreview.net/forum?id=1RfgHzf5IA*

## Abstract

Structure observed in trained recurrent networks may be inherited from input encodings rather than learned from data. We develop and apply a three-step decomposition to disentangle the two: (1) compare trained representations against untrained baselines to isolate input-driven structure, (2) compare against information-theoretic bounds to quantify what is achievable without learning, and (3) use causal interventions to test whether inherited and learned components are functionally used. Applied to GRUs trained via behavioral cloning on aliased navigation in a 127-node binary tree, the most prominent hidden-state feature, a depth gradient on PC1, is already present before training: an untrained GRU captures 97% of the trained correlation, reflecting input structure rather than learned spatial knowledge. What training adds is within-class node discrimination via sequential memory. Replacing depth-stratified observations with random class assignments eliminates the inherited axis; the GRU compensates with $7\times$ greater learned spatial discrimination while maintaining comparable performance. PCA ablation reveals a double dissociation in exploration pattern, confirming that both inherited and learned components are causally involved in behavior. Applied to a non-hierarchical radial arm maze, the framework recovers an analogous inherited axis but qualitatively different learned structure: visit history tracking rather than spatial disambiguation.

## 1 Introduction

Suppose you train a gated recurrent unit (GRU) to navigate a maze under partial observability, examine its hidden states, and find that the first principal component correlates strongly with the agent's depth in the maze ($\rho = 0.89$). The natural conclusion is that training drove the network to build a spatial representation of depth. But an untrained GRU, with random weights and no exposure to trajectory data, shows nearly the same correlation ($\rho = 0.87$). The structure was inherited from the input encoding, not learned.

A growing literature characterizes structured representations in trained recurrent networks: grid cells in path integration networks (Banino et al., 2018; Cueva & Wei, 2018), place-like responses in navigation agents (Whittington et al., 2020), interpretable features in language models (Karpathy et al., 2015), but typically without controlling for what the input encoding already provides. If an encoding stratifies observations by some feature, any network processing those inputs will reflect that stratification in its hidden states regardless of training. This connects to a broader phenomenon: Lampinen et al. (2024) show that simpler features systematically explain more representation variance, including dominating top principal components, even when all features are computed equally well, implying that the most visible structure in a trained network may reflect input properties rather than what training contributed. Reporting this as "the network learned to encode [feature]" overstates what training contributed.

We investigate this using GRUs trained to imitate mouse navigation in the binary tree labyrinth of Rosenberg et al. (2021), an environment that creates an extreme case of structural aliasing: 127 maze nodes produce only 5 unique observations under a local structural encoding, yielding a 25:1 aliasing ratio. The encoding

is depth-stratified by construction, with each observation class corresponding to a contiguous band of tree depths, creating a strong prior expectation that any depth structure in hidden states could be inherited rather than learned. At the same time, navigating this maze requires disambiguating perceptually identical locations using memory, making it a natural testbed for separating inherited from learned representations. Our primary contribution is a decomposition combining untrained baselines, information-theoretic bounds, and causal interventions, developed here for recurrent navigation agents with known input encodings. The decomposition is most precise when the input encoding is fixed and known, as in designed observation models or pretrained frozen encoders; extending it to end-to-end learned perceptual encodings would require defining the untrained baseline over the full processing pipeline, including the untrained encoder.

In this environment, recurrence adds only a thin behavioral margin over the memoryless baseline, a feature, not a limitation, of our testbed. When a recurrent network far outperforms its memoryless baseline, the learned contribution is self-evident from behavior alone; when the margin is thin yet the internal representations are rich, attributing that richness correctly requires distinguishing what was inherited from what was learned. The decomposition is most revealing precisely in this regime.

Our core finding is that equivalent behavior can arise from qualitatively different inherited/learned partitions, and that the partition is determined by the input encoding. We establish this through a three-step decomposition: untrained baselines (to isolate input contributions), Bayesian filtering bounds (to bracket what is achievable with and without action observation), and causal PCA ablation (to test functional use), with linear probing as the measurement tool throughout. GRUs are trained via cross-entropy behavioral cloning on real mouse trajectory data; for a T-maze control, we use Maximum Causal Entropy Inverse Reinforcement Learning (IRL) (Ziebart et al., 2008) to derive soft policy targets. Neither training method is our contribution; we treat the trained network as an object of scientific study.

In the binary tree, the dominant hidden-state geometry is inherited: an untrained GRU accounts for 97% of the PC1–depth correlation (Section 3.2). What training adds is within-class node discrimination orthogonal to the inherited axis. A controlled encoding swap confirms this partition is encoding-determined: replacing depth-stratified observations with random class assignments eliminates the inherited axis while the GRU compensates with $7\times$ greater learned discrimination and near-equivalent performance (Section 3.3). PCA ablation reveals a double dissociation in exploration pattern, confirming both components are causally used (Section 3.4). Applied to a non-hierarchical radial arm maze, the framework recovers an analogous inherited axis but qualitatively different learned structure: visit history tracking rather than spatial disambiguation (Section 3.5). These findings hold across architectures and are task-specific rather than architectural defaults (Sections D–E).

## 2 Methods

### 2.1 Environment and Data

The maze is a complete binary tree of depth 6, comprising 63 internal branch nodes and 64 terminal leaf nodes for a total of 127 nodes (Rosenberg et al., 2021). Mice enter at the root (node 0) and navigate toward a water port at node 100, located at depth 6. At each node the mouse selects one of three actions: descend to the left child, descend to the right child, or reverse toward the parent. At leaves, the two descend actions produce self-loops; at the root, reversing is a self-loop. Transitions are fully deterministic.

We use behavioral data from two sources derived from the same experimental cohort of 10 water-restricted mice.

**Rosenberg raw data (primary).** We load the original trajectory recordings from Rosenberg et al. (2021), strip the exit marker (node 127), deduplicate consecutive identical nodes (collapsing implicit stays), and infer actions from consecutive node pairs. This yields 1,578 variable-length bouts split 80/20 into 1,263 training and 315 validation bouts. Unlike the processed dataset used in prior IRL work on this maze (Ashwood et al., 2022), the raw trajectories visit all 127 nodes and preserve the full behavioral variability of the animals.

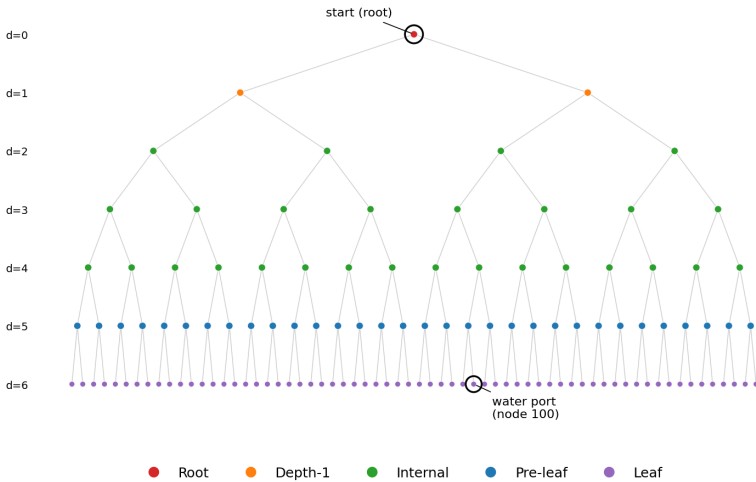

Figure 1: **Binary tree labyrinth** (Rosenberg et al., 2021). 127 nodes, depth 6. The mouse enters at the root and navigates toward the water port at node 100. Node colors indicate the 5 observation classes produced by the structural encoding (Table 1): root, depth-1, internal (depths 2–4), pre-leaf (depth 5), and leaf (depth 6); 25:1 aliasing ratio.

**Structural observation encoding.** To create partial observability, we encode each node as a 12-dimensional vector capturing only local tree structure: the current node's degree (normalized by 3), binary root and leaf flags, and the same three features for each of the three action destinations (left child, right child, parent). The agent's own actions are not part of the input; action history must be inferred from observation context. This encoding produces exactly 5 unique observation classes across all 127 nodes (Table 1), creating a 25:1 aliasing ratio. For example, all 64 leaves produce identical observations, as do all 28 nodes at depths 2–4. A memoryless policy can learn only one shared action distribution per observation class; any finer-grained spatial behavior requires sequential memory.

Table 1: Structural observation classes. The 12-dimensional encoding produces only 5 unique observations across 127 nodes.

| Class | Depths | Nodes | Distinguishing feature |
|---|---|---|---|
| Root | 0 | 1 | Degree 2, children are internal |
| Depth 1 | 1 | 2 | Parent is root |
| Internal | 2–4 | 28 | All neighbors have degree 3 |
| Pre-leaf | 5 | 32 | Children are leaves |
| Leaf | 6 | 64 | Degree 1, descend actions are self-loops |

This contrasts with the $k$-hop binary mask approach used in prior work, where a 127-dimensional vector provides a nearly unique fingerprint at each position, allowing even a memoryless MLP to match the full-information ceiling. The structural encoding eliminates this confound: with only 5 distinct inputs, any GRU advantage is directly attributable to sequential memory.

**Random observation encoding.** As a control, we also train networks under a random encoding variant that assigns nodes to 5 observation classes at random, destroying any correlation with tree depth while preserving the 25:1 aliasing ratio and observation dimensionality. Full specification is in Section 3.3.

## 2.2 Training Pipeline

### 2.2.1 Behavioral Cloning (Labyrinth Experiments)

A GRU (Cho et al., 2014) is trained to reproduce mouse navigation behavior from aliased structural observations via cross-entropy minimization. The architecture processes each 12-dimensional observation through a ReLU embedding layer, a single-layer GRU with hidden dimension $d = 128$ (default), and a two-layer policy head with ReLU activation and softmax output. Dropout ($p = 0.1$) is applied between the GRU and the policy head.

The training objective is the cross-entropy between the GRU's predicted action distribution and Laplace-smoothed empirical action frequencies $\hat{\pi}(a \mid s)$ computed directly from the Rosenberg trajectory data (pseudocount = 1):

$$\mathcal{L}(\theta) = -\frac{1}{T} \sum_{t=1}^{T} \sum_{a} \hat{\pi}(a \mid s_t) \log \pi_\theta(a \mid \mathbf{h}_t). \tag{1}$$

Sequences longer than 200 steps are split into non-overlapping chunks with the hidden state detached at chunk boundaries. All experiments use 5 random seeds unless otherwise noted; remaining training hyperparameters are reported in Appendix A.

### 2.2.2 IRL-Based Distillation (T-Maze Experiment)

For the T-maze control (Section E), we use Maximum Causal Entropy IRL (Ziebart et al., 2008; Ziebart, 2010) to recover a reward function and derive soft policy targets. The GRU is then trained via the same cross-entropy objective (Eq. 1), with the IRL-derived soft policy replacing empirical frequencies. Details are in Appendix A.

**Memoryless baseline.** An MLP with identical architecture minus the GRU layer serves as the memoryless baseline: it processes each observation independently through the same embedding, hidden layer, and policy head, with no recurrent state carried across timesteps. Any performance gap between GRU and MLP is therefore directly attributable to sequential memory.

## 2.3 Analysis Methods

We characterize the GRU's learned representations using four complementary approaches.

### 2.3.1 Representational Similarity Analysis (RSA)

Following Kriegeskorte et al. (2008), we construct a representational dissimilarity matrix (RDM) from pairwise Euclidean distances between mean hidden states at each of the 127 nodes, then compare this neural RDM against four model RDMs capturing different spatial hypotheses: depth difference (absolute difference in tree depth), graph distance (shortest-path length on the tree), path Hamming distance (number of differing left/right decisions from root), and subtree identity (binary: same or different depth-2 subtree). We quantify correspondence using Mantel tests with Spearman correlation and 9,999 permutations.

### 2.3.2 Linear Probing

We train a 127-way logistic regression classifier (Alain & Bengio, 2016) on GRU hidden states to predict the true node identity, using the same train/validation split as the policy. This assesses how much location information is linearly accessible in the hidden state. We report overall accuracy and per-observation-class breakdowns, comparing against chance levels that account for class imbalance ($1/127 \approx 0.8\%$ overall; $1/64 \approx 1.6\%$ within leaves).

### 2.3.3 Causal PCA Ablation

To move beyond correlational analysis, we perform causal interventions on the hidden state. We compute the principal components of hidden states across the validation set, then ablate specific components by projecting

them out of the hidden state at every timestep during forward evaluation. Critically, the projection is applied both to the policy output *and* to the recurrent state passed to the next GRU step. This creates a propagating "lesion" where the ablated information is destroyed, not merely hidden from the output layer. We measure the impact on policy log-likelihood and per-node action distributions (quantified as total variation distance from the intact policy). A random-direction control, projecting out a single random unit-norm vector, provides a baseline for the expected effect of removing one dimension of variance.

### 2.3.4 Bayesian Belief Filters

We compare the GRU against three Bayesian belief filter variants. The **factorized memoryless-action filter** maintains the posterior $p(s_t \mid o_{1:t})$ under the assumption that actions are random at each step, with transitions marginalized over the behavioral cloning policy. The **joint-state filter with first-order action smoothing** maintains a posterior over (state, previous-action) pairs under the empirical action policy $\pi(a \mid s, a_{t-1})$; the factorized variant cannot represent this action history information, which the GRU may infer from observation context. The **action-informed** filter is given the true action at each step and provides an upper bound on location tracking under this encoding. All three process the same 200-step chunks as the GRU, resetting at chunk boundaries.

For all paired comparisons we report BCa bootstrap 95% confidence intervals (Efron, 1987) and Hedges' $g$ effect sizes. With $n = 5$ seeds the minimum achievable sign-flip permutation $p$-value is $1/16$, so we rely primarily on effect sizes rather than $p$-values.

## 2.4 Radial Arm Maze Environment

To test whether the decomposition framework generalizes beyond tree-structured environments, we apply it to an 8-arm radial maze, a star graph with fundamentally non-hierarchical topology (Olton & Samuelson, 1976). The maze comprises a central hub (node 0) connected to 8 arms of 3 nodes each, for a total of 25 nodes. Each arm tip contains a reward that depletes after the agent's first visit. The task is to visit all 8 tips without unnecessary revisits, requiring the agent to track which arms have been explored, a form of episodic memory qualitatively distinct from the spatial disambiguation required in the binary tree.

**Actions.** The agent has 10 possible actions: 8 arm-entry actions at the hub and 2 corridor actions (advance toward tip, retreat toward hub) within arms. At tips, advance produces a self-loop. Invalid actions at each node produce self-loops.

**Observation encoding.** A 12-dimensional structural encoding produces 4 observation classes (hub, proximal, medial, tip) with a 6.25:1 aliasing ratio: all 8 tips are perceptually identical, as are all 8 proximal and all 8 medial nodes. The encoding is perfectly correlated with radial distance from the hub (0–3), creating the same prior expectation as the tree: any radial-distance structure in hidden states could be inherited. A random encoding variant assigns nodes to 4 classes at random, destroying the radial-distance correlation.

**Trajectory generation.** We generate 1,500 synthetic trajectories from a noisy-optimal policy (0.8 probability of optimal action at each step). The GRU is trained via the same behavioral cloning objective (Eq. 1), with identical architecture ($d = 128$), training procedure, and analysis pipeline as the binary tree experiments.

## 3 Results

### 3.1 Performance Under Aliasing and Bayesian Bounds

Table 2 summarizes performance under structural observation aliasing. The GRU–MLP gap is $+0.011$ bits/dec, consistent across all seeds, closing 27% of the gap to the full-information ceiling.

---

[1]Standard deviations below 0.0005 bits/dec are rounded to 0.000. The actual cross-seed standard deviation for GRU log-likelihood is $\approx 0.0002$ bits/dec.

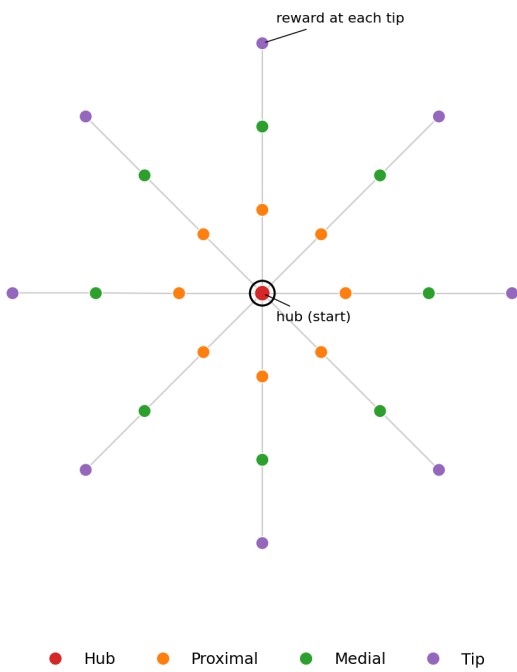

Figure 2: **Radial arm maze** (Olton & Samuelson, 1976). 25 nodes: a central hub connected to 8 arms of 3 nodes each. Node colors indicate the 4 observation classes (hub, proximal, medial, tip); the 8 tips, 8 proximal, and 8 medial nodes are perceptually identical within each class. Reward depletes after first visit at each tip.

Table 2: Performance comparison under structural observation aliasing (5 unique observations for 127 nodes). Log-likelihood in bits per decision (higher is better). All values are mean $\pm$ std across 5 random seeds.[1]

| Model | LL (bits/dec) | Top-1 Accuracy |
|---|---|---|
| Random policy | $-1.585$ | 33.3% |
| MLP (memoryless) | $-1.281 \pm 0.000$ | $51.5\% \pm 0.0\%$ |
| **GRU** | $\mathbf{-1.270 \pm 0.000}$ | $\mathbf{51.9\% \pm 0.2\%}$ |
| BC ceiling (full state) | $-1.239$ | — |

The GRU's linear probe achieves 25.3% node decoding accuracy, between the factorized memoryless-action filter (22.7%) and the joint-state filter with first-order action smoothing (29.3%; Table 3). The action-informed filter reaches 94.4%, confirming that action uncertainty is the primary bottleneck for location tracking under aliasing.

**Action prediction probes.** The +2.6 percentage-point gap between the GRU (25.3%) and the factorized filter (22.7%) reflects the value of action history information that the factorized filter cannot represent. We test this interpretation directly by training a linear probe on GRU hidden states to predict the previous action. The trained GRU achieves 78.2%, above both the observation-transition baseline (71.3%, which captures what a single $(o_{t-1}, o_t)$ pair reveals) and the untrained GRU (70.9%). The +6.97 percentage-point gap (95% paired BCa bootstrap CI across 5 seeds, 9999 resamples: $[+6.44, +7.52]$) is consistent with the GRU partially recovering, from observation context alone, action-history information that the joint-state filter consumes directly. The per-action breakdown localizes this gain. Reverse actions are trivially inferable from any single transition ($\sim$100% for all methods), but distinguishing left from right requires knowing

Table 3: Node decoding accuracy and policy quality across models and Bayesian filter variants. The GRU sits between the factorized filter (22.7%) and the joint-state filter with first-order action smoothing (29.3%).

| Model | Node Acc | LL (bits/dec) | Gap Closed |
|---|---|---|---|
| No memory (obs class) | 7.1% | −1.283 | — |
| MLP (trained) | — | −1.281 | 0% |
| Factorized memoryless-action filter | 22.7% | −1.278 | 7% |
| **GRU (trained)** | **25.3%** | **−1.270** | **27%** |
| Joint-state filter (action smoothing) | 29.3% | — | — |
| Action-informed filter | 94.4% | −1.241 | 95% |
| BC ceiling (full state) | 100% | −1.239 | 100% |

which subtree the agent occupies, and it is on left actions that the trained GRU achieves 66.6% where the observation-transition baseline sits near chance ($\sim$42%).

Ablation analysis (Section 3.4) shows that PC1 (depth) carries a behaviorally large causal effect ($-0.21$ bits/dec when removed, well outside the 95% range of a 1000-draw random-direction null; Table 7).

## 3.2 Decomposing Inherited and Learned Structure

RSA reveals depth-dominated hidden-state geometry (Figure 4): depth difference is the strongest correlate ($\rho = 0.858$), followed by path Hamming distance ($\rho = 0.451$), while subtree identity is non-significant ($\rho = 0.009$). Graph distance is *negatively* correlated ($\rho = -0.246$), indicating that nodes at the same depth are close in hidden-state space despite being far apart on the tree. This geometry is consistent with the input encoding, which maps 127 nodes to 5 depth-stratified observation classes; the within-class probe accuracy (below) is what demonstrates learned encoding beyond input structure.

PCA of the 128-dimensional hidden states shows that 4 components capture 81% of variance, with PC1 alone at 46.8%. PC1 correlates strongly with tree depth ($\rho = 0.890 \pm 0.044$ across 5 seeds), but an untrained GRU achieves nearly the same ($\rho = 0.865 \pm 0.026$; Table 5). The trained$-$untrained difference is small ($+0.026$, paired BCa 95% CI $[-0.007, +0.063]$, crossing zero), so this geometry reflects the input encoding rather than learned spatial knowledge, with the inheritance ratio at 97% (paired bootstrap 95% CI $[93\%, 101\%]$). Training's contribution appears not in PC1$-$depth correlation but in the probe accuracy gap, which rises from 15.0% on raw observations to 25.7% on trained hidden states (paired BCa 95% CI on the gap over the trajectory-shuffle floor: $[+10.6, +12.4]$pp). PCs 2–4 carry weaker positional signals and are associated with subtree and directional information (Section 3.4).

A 127-way linear probe achieves 25.3% accuracy ($32\times$ chance; Table 4), with the strongest relative performance among the 64 identical leaves (15.4%, or $9.6\times$ chance). A nonlinear probe ($26.3\% \pm 0.8\%$) adds negligibly, confirming that the bottleneck is representational rather than in the readout, consistent with the Bayesian filter analysis (Table 3).

**Framing against the theoretical ceiling.** The GRU captures roughly 13% of the probe-accuracy gap between raw observations (15.0%) and the action-informed ceiling (94.4%), distinct from the 27% log-likelihood gap closed (Table 2). This modest fraction reflects the environment's high behavioral stochasticity (mean policy entropy of 0.79 out of 1.58 bits), which severely limits what any system can infer without directly observing actions.

**Probe baselines: decomposing input structure from learned encoding.** Table 5 compares probe accuracy and PC1–depth correlation across four baselines that isolate the contributions of input encoding, recurrent architecture, and training.

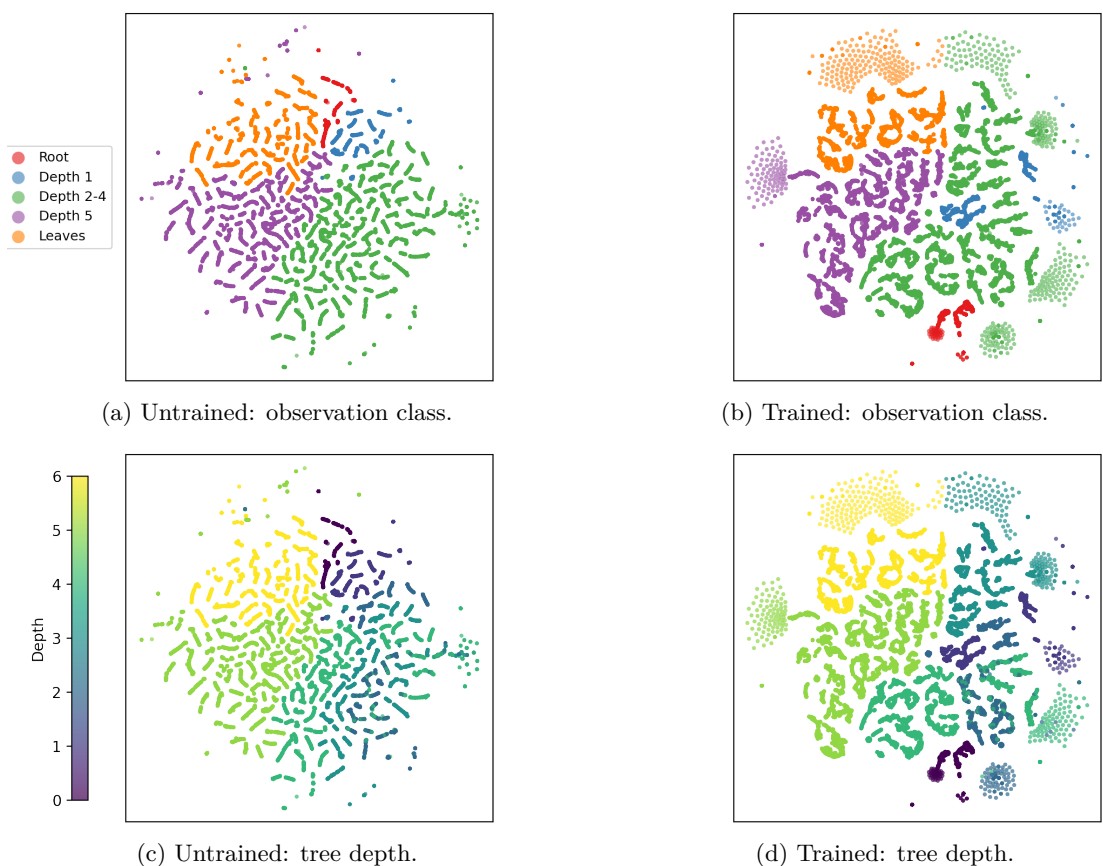

(a) Untrained: observation class.

(b) Trained: observation class.

(c) Untrained: tree depth.

(d) Trained: tree depth.

Figure 3: **Untrained vs. trained hidden-state geometry.** t-SNE of GRU hidden states before and after training. Top row: colored by observation class. Bottom row: colored by tree depth. The untrained network already clusters by observation class (reflecting input structure), but the trained network develops finer within-class subclusters corresponding to individual nodes.

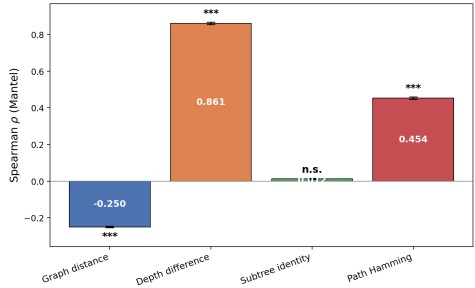

Figure 4: **RSA reveals depth-dominated hidden-state geometry.** Mantel test Spearman correlations between the neural RDM and four spatial model RDMs, with 95% CIs. Bar heights show per-seed means ($\rho = 0.861, -0.250, 0.454$); text reports aggregate Mantel test statistics computed on the pooled RDM ($\rho = 0.858, -0.246, 0.451$).

The raw observation probe (15.0%) and untrained GRU (15.8%) are nearly identical, showing the recurrent architecture contributes nothing before training. The two trajectory aware selectivity controls, trajectory-

Table 4: Linear probe accuracy by observation class. The GRU disambiguates nodes within each aliased group at well above chance, with the strongest relative performance among the 64 identical leaves.

| Obs Class | Nodes | Probe Acc | Chance | Over Chance |
|---|---|---|---|---|
| Root | 1 | 100.0% | 100.0% | 1.0× |
| Depth 1 | 2 | 60.8% | 50.0% | 1.2× |
| Depths 2–4 | 28 | 27.4% | 3.6% | 7.6× |
| Depth 5 | 32 | 15.4% | 3.1% | 5.0× |
| Leaves | 64 | 15.4% | 1.6% | 9.6× |
| **Overall** | **127** | **25.3%** | **0.8%** | **32×** |

Table 5: Probe baselines decomposing input structure from learned spatial encoding. Two trajectory aware selectivity controls give the chance floor: *traj* swaps every held-out trajectory's labels with those of another random trajectory, and *class* permutes labels only within each observation class. We also report an i.i.d. shuffle, which under-states the floor because it preserves each trajectory's marginal node-visit frequency for the probe to latch onto (Hewitt and Liang 2019). The trajectory aware floors are about 15%, and the learned contribution is the trained GRU's gain over those floors and over raw observations ($\sim$+11 pp). Mean ± std across 5 seeds.

| Baseline | Probe Acc | \|PC1–depth\| $\rho$ | \|PC1–time\| $\rho$ |
|---|---|---|---|
| Raw observation | 15.0% | — | — |
| Untrained GRU | 15.8% ± 0.2% | 0.865 ± 0.023 | 0.163 ± 0.018 |
| Selectivity (i.i.d., diagnostic) | 4.6% ± 0.1% | — | — |
| Selectivity (trajectory) | 14.4% ± 0.5% | — | — |
| Selectivity (within-class) | 14.7% ± 0.6% | — | — |
| **Trained GRU** | **25.7% ± 0.7%** | **0.890 ± 0.039** | **0.103 ± 0.047** |
| Trained GRU (depth-regressed) | 26.0% ± 0.6% | — | — |
| MLP probe (nonlinear) | 26.3% ± 0.8% | — | — |
| Chance | 0.8% | — | — |

level shuffle (14.4%) and within-observation-class shuffle (14.7%), give the relevant chance floor: this is what the probe extracts when label structure is destroyed but observation-conditional statistics are preserved. The i.i.d. shuffle (4.6%, retained as a diagnostic only) under-states that floor by a factor of $\sim$3×, because it preserves each trajectory's marginal node-visit frequency for the probe to memorize. The trained GRU's +11 percentage-point advantage over both raw observations and the trajectory aware selectivity floors, i.e., above-and-beyond what is achievable from observation-class marginals alone, represents learned within-class node discrimination: the GRU telling apart nodes that produce identical observations, using sequential memory.

**Depth-regression analysis.** To test whether the learned spatial encoding operates through the depth axis, we fit a linear regression from hidden states to tree depth, project out the resulting direction (reducing the depth $R^2$ to approximately zero), and retrain the 127-way node probe on the residuals. Removing all linear depth information from the trained GRU's hidden states leaves node discrimination essentially intact (26.0% ± 0.6% vs. 25.7% ± 0.7%; Table 5), confirming that the learned spatial encoding is largely orthogonal to depth.

t-SNE embeddings (Figure 3) confirm this visually: the untrained network already clusters by observation class, while training fragments these clusters into node-level subclusters, demonstrating that the GRU maps identical inputs to different internal states depending on action history.

**Supplementary analyses.** The inherited depth axis is present from the first timestep of each episode and is stable throughout (Appendix C), consistent with reflecting input structure rather than accumulating over time. Across hidden dimensions from 4 to 256, PC1–depth $\rho$ remains high while probe accuracy scales monotonically beyond where log-likelihood saturates ($d \approx 32$; Appendix B), indicating that training builds richer spatial representations than the behavioral objective strictly requires.

### 3.3 Encoding Swap: The Inherited/Learned Partition is Encoding-Determined

**Random encoding specification.** To test whether the decomposition is specific to the depth-stratified structural encoding, we construct a random encoding variant. Nodes are assigned uniformly at random to 5 observation classes (sizes 26, 26, 25, 25, 25), each given a distinct random 12-dimensional observation vector. This preserves the 25:1 aliasing ratio and observation dimensionality while destroying any correlation between observation class and tree depth (Spearman $\rho = 0.085$ for the assignment used, vs. $\approx 0.7$ for the structural encoding). A single random class assignment is used across all 5 training seeds; the training pipeline, GRU architecture, and analysis methods are identical.

Table 6 presents the $2 \times 2$ comparison. The two encodings produce qualitatively opposite inherited/learned patterns while converging on similar behavioral performance.

Table 6: Encoding swap comparison across structural and random observation classes, before and after training. PC1–depth $\rho$ captures the inherited depth axis; probe accuracy captures learned spatial discrimination (balanced accuracy for random encoding; see text). Val LL in bits per decision. Random encoding values are mean $\pm$ std across 5 seeds; structural values are from Section 3.2.

| Encoding | Condition | PC1–depth $\rho$ | Probe Acc | Val LL |
|---|---|---|---|---|
| Structural | Untrained | 0.865 | 15.8% | — |
| | Trained | 0.890 | 25.7% | $-1.270$ |
| Random | Untrained | $0.070 \pm 0.067$ | $13.3\% \pm 0.1\%$ | — |
| | Trained | $0.026 \pm 0.015$ | $80.9\% \pm 0.5\%$ | $-1.288 \pm 0.001$ |

**The inherited component toggles cleanly.** Under structural encoding, the untrained GRU shows strong depth organization ($\rho = 0.865$); under random encoding, $\rho = 0.070$ (noise). Training preserves this difference (Table 6). The trained GRU does not recover the depth axis when the encoding fails to provide it, confirming that depth geometry is donated by the input rather than driven by the behavioral cloning objective.

**The learned component inverts.** Under structural encoding, training adds +11 percentage points of probe accuracy. Under random encoding, training adds +77.0 percentage points ($3.9\% \rightarrow 80.9\%$ over the memoryless raw-observation baseline; equivalently +67.6 pp over the untrained-GRU baseline of 13.3% reported in Table 6). When the encoding donates less inherited structure, the network must learn more, and it does so substantially.

**Behavioral performance is nearly matched.** Despite fundamentally different internal organization, the two encodings reach similar log-likelihoods (0.018 bits/dec gap; Table 6), demonstrating that equivalent input–output behavior can arise from qualitatively different representational strategies.

**Two distinct mechanisms underlie the swap.** The inherited component tracks the encoding's correlation with spatial structure: when observations correlate with depth, hidden states inherit a depth axis; when they do not, no depth axis appears. The learned component is modulated by a separate property of the encoding: transition diversity. Depth-stratified classes produce degenerate transition signatures, severely

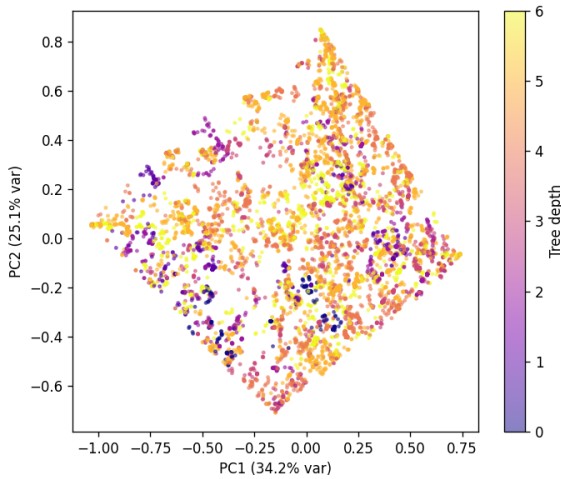 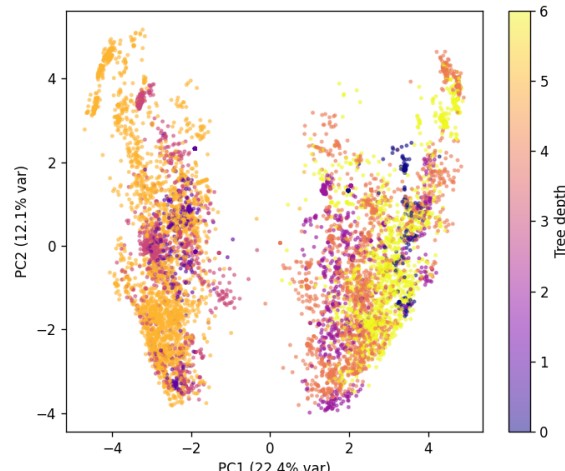

(a) Untrained GRU: no depth gradient ($\rho = 0.07$).

(b) Trained GRU: rich node-level structure (80.9% probe accuracy), no depth gradient ($\rho = 0.03$).

Figure 5: **PCA of GRU hidden states under random encoding, colored by tree depth.** Left: the untrained GRU shows no depth gradient, confirming the structural encoding's depth geometry is inherited rather than architectural. Right: training produces rich node-level spatial structure without a depth axis, in direct contrast to the depth-dominated geometry under structural encoding (Figure 3). PC1 variance explained: $32.5\% \pm 2.0\%$ (untrained) and $22.3\% \pm 0.7\%$ (trained).

limiting what sequential memory can distinguish.[2] Random classes, distributed across all tree depths, yield near-maximal transition diversity (74 of 75 triples), which the GRU exploits for 7$\times$ greater learned discrimination. These are two distinct mechanisms: the inherited component depends on the encoding's spatial correlation structure, while the learned component depends on the encoding's transition informativeness. The swap changes both simultaneously. The behavioral near-equivalence despite opposite internal organization demonstrates that the network adapts its inherited/learned partition to whatever the encoding provides.

**The decomposition changes the interpretation.** Without the untrained baseline, a trained-only analysis would attribute the $\rho = 0.890$ depth correlation under structural encoding to learning, when 97% is inherited. Under random encoding, the 80.9% probe accuracy is correctly identified as learned (the untrained baseline of 13.3% confirms this). The decomposition reveals these as different inherited/learned partitions rather than different network capabilities.

Random encoding probe results use balanced accuracy to account for unequal class sizes; the qualitative conclusion is robust to this choice.[3]

### 3.4 Causal Evidence: A Depth–Direction Double Dissociation in Exploration Pattern

Correlational analyses establish that the GRU encodes depth and subtree information, but do not show whether these encodings are causally involved in behavior. We address this with PCA ablation experiments that remove specific variance directions from the hidden state (Section 2.3).

**Global impact.** Removing PC1 (depth) drops log-likelihood by $-0.210\,\text{bits/dec}$, eliminating 67% of the GRU's advantage over random (Table 7). The null distribution of 1000 random unit-norm direction ablations

---

[2]Under structural encoding, only 18 of 75 possible ($\text{obs}_t$, action, $\text{obs}_{t+1}$) triples occur and only 1 of 127 nodes is uniquely identifiable from a single transition. Under random encoding, 74 of 75 triples occur and 64 of 127 nodes (50.4%) are uniquely identifiable from one step.

[3]Structural encoding results use unbalanced accuracy (Table 4). The direction of the effect (random $\gg$ structural on the learned gain) holds under either metric.

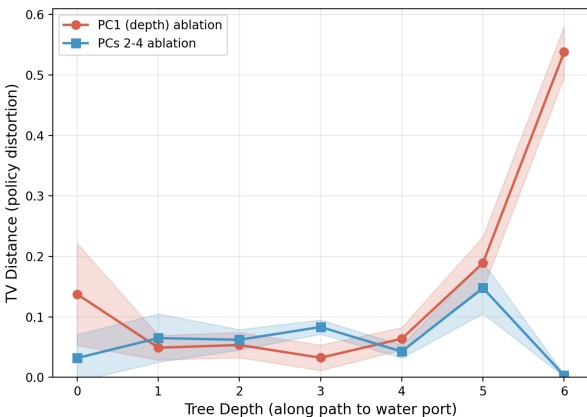

Figure 6: **Spatial double dissociation.** Policy distortion (TV distance from intact policy) along the root-to-water path for PC1 (depth) vs. PCs 2–4 (direction) ablation. PC1 ablation peaks at leaves (the network forgets it is at a dead end); PCs 2–4 ablation peaks at mid-tree branching points (the network loses subtree discrimination). The profiles cross between depths 3 and 4. Mean $\pm$ 1 std across 5 seeds.

has mean $-1.270$ and standard deviation $0.0013$ bits/dec (95% range $[-1.273, -1.267]$). PC1's effect lies far outside this null, and even PC4's small effect ($-0.009$ bits/dec) lies outside the 95% range, indicating that all five PC ablations are reliably distinguishable from random unit-direction projections. PCs 5–50 together cost only $-0.028$ bits/dec, confirming that behaviorally relevant information concentrates in the first few components.

Table 7: Policy log-likelihood under PCA ablation, ranked against a 1000-draw unit-norm random-direction null (seed 1234; null mean $-1.270$, SD $0.0013$ bits/dec, 95% range $[-1.273, -1.267]$). The Deviation column reports the magnitude of each ablation's effect in null-SD units; the null is direction-matched rather than variance-matched (each random unit direction captures $O(1/d)$ of hidden-state variance whereas PC1 captures 46.8%), so this column quantifies how each ablation compares to a typical random direction of the same norm, not a variance-matched effect size. All five PC ablations sit outside the empirical 95% null range, including PC4.

| Ablation | LL (bits/dec) | $\Delta$ LL | Deviation (null-SD) |
|---|---|---|---|
| No ablation | $-1.270$ | — | — |
| Ablate PC1 (depth) | $-1.480$ | $-0.210$ | $-158.5$ |
| Ablate PC2 | $-1.316$ | $-0.047$ | $-34.8$ |
| Ablate PC3 | $-1.302$ | $-0.032$ | $-24.1$ |
| Ablate PC4 | $-1.279$ | $-0.009$ | $-6.8$ |
| Ablate PCs 2–4 | $-1.340$ | $-0.071$ | $-53.1$ |
| Ablate PCs 5–50 | $-1.297$ | $-0.028$ | $-20.7$ |
| Random direction (1000 draws) | $-1.270 \pm 0.001$ | $\sim 0$ | — |
| Random baseline | $-1.585$ | — | — |

**Spatial dissociation.** Per-node analysis reveals a functional double dissociation (Figure 6). PC1 ablation produces its largest distortion at leaves (TV $= 0.54$): the intact policy assigns $\sim$99% probability to reversing, but under ablation this degrades to $\sim$50% reverse, 25% left, 25% right. The network *forgets it is at a leaf.* PCs 2–4 ablation produces the complementary deficit: distortion peaks at mid-tree branching points where subtree discrimination matters and drops to near-zero at leaves. The two profiles cross between depths 3 and 4.

**Behavioral rollouts.** Behavioral rollouts ($n = 2000$ trajectories per condition per seed, 5 GRU seeds; Table 8) corroborate the per-node TV-distance pattern. Both ablations reduce water-finding compared with the intact policy ($19.1\% \pm 0.4$ across seeds). On the cross-seed paired test PC1 ablation impairs water-finding more than PCs 2–4 ablation ($12.2\% \pm 2.3$ vs. $16.6\% \pm 2.4$ across seeds; paired BCa 95% CI on the (PC1 − PCs 2–4) difference $[-6.8, -0.9]$ pp, excluding zero), though per-seed variability is substantial. The (PC1 − PCs 2–4) gap ranges from $-7.1$ to $+1.2$ pp across the 5 seeds, so a single-seed analysis can flip the direction and the cross-seed paired test is the appropriate level of inference. The exploration-pattern dissociation, in contrast, is robust in every seed individually. PC1 ablation drives trajectories deeper into the tree (mean depth $4.31 \pm 0.42$ vs. $3.97$ baseline) while reducing the number of unique nodes visited in every seed ($35.2$ baseline $\rightarrow 24.9$–$32.7$ under PC1 ablation), the pattern expected when the network forgets it is at a leaf and oscillates in the deep portion of the tree. PCs 2–4 ablation produces the complementary signature, mean depth near baseline ($3.96 \pm 0.14$) with breadth intact or slightly increased ($36.2 \pm 1.0$ unique nodes, range $34.5$–$37.0$ across seeds), as the network retains depth awareness but loses left–right discrimination at branching points.

Table 8: Behavioral rollouts under PCA ablation across 5 GRU seeds ($n = 2000$ trajectories per condition per seed). Reach-water rates, mean depth, and unique nodes visited are mean $\pm$ 1 SD across seeds. The paired BCa 95% CI is on the seed-matched (PC1 − PCs 2–4) reach-water difference (9999 resamples). PC1 ablation impairs water-finding more than PCs 2–4 ablation and reduces exploration breadth (unique nodes visited) in every seed; PCs 2–4 ablation leaves breadth intact.

| Condition | Reach water | Mean depth | Unique nodes |
|---|---|---|---|
| No ablation | $19.1\% \pm 0.4$ | $3.97$ | $35.2 \pm 0.1$ |
| Ablate PC1 | $12.2\% \pm 2.3$ | $4.31 \pm 0.42$ | $29.7 \pm 3.1$ |
| Ablate PCs 2–4 | $16.6\% \pm 2.4$ | $3.96 \pm 0.14$ | $36.2 \pm 1.0$ |
| Paired BCa 95% CI on (PC1 − PCs 2–4) reach water: $[-6.8, -0.9]$ pp. | | | |

**Training dynamics.** Training dynamics confirm gradual accumulation of spatial encoding with no phase transition; the within/between-class distance ratio shows compression then expansion consistent with the depth-regression analysis (Appendix C).

**Architecture comparison.** Across four architectures (GRU, LSTM, GTrXL, minGRU), behavioral performance is equivalent but representational geometry diverges: gated recurrence concentrates depth on PC1 while attention distributes it across dimensions, confirming the depth-first geometry is architecture-specific rather than task-required (Appendix D).

**Task specificity.** A T-maze control confirms the geometry is task-specific: the GRU encodes the one bit the task demands rather than a depth gradient, complementing the encoding swap's demonstration that the geometry is also encoding-specific (Appendix E).

### 3.5 Radial Arm Maze: Different Task, Different Learned Structure

To test generality beyond tree-structured environments, we apply the identical framework to an 8-arm radial maze (Section 2.4), a star graph requiring episodic visit memory. The GRU–MLP gap ($+0.165$ bits/dec) is $15\times$ larger than in the binary tree, reflecting that foraging fundamentally requires visit tracking that a memoryless policy cannot provide.

The inherited/learned decomposition replicates in structure but diverges in content. The untrained GRU's PC1 correlates with radial distance ($|\rho| \approx 0.82$), and training barely changes this (Table 9), paralleling the binary tree. But the learned component is categorically different: the node probe is flat ($27.2\% \rightarrow 27.4\%$), while visit-bit and $N$-visited probes show large trained gains (Table 9). The GRU learns to track which arms have been explored, a form of episodic memory with no untrained counterpart, rather than to discriminate

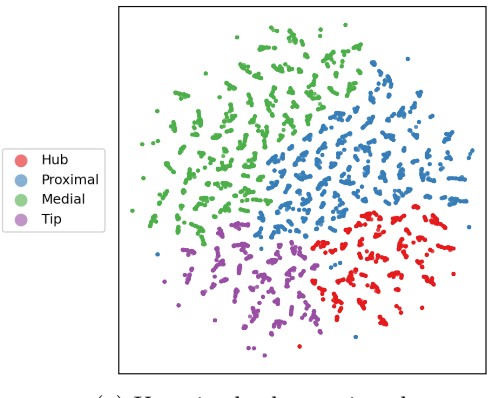

(a) Untrained: observation class.

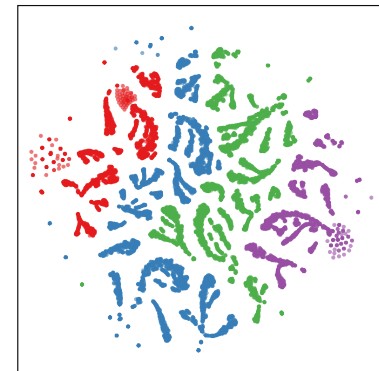

(b) Trained: observation class.

Figure 7: **Untrained vs. trained hidden-state geometry (radial arm maze).** t-SNE of GRU hidden states colored by observation class. (a) The untrained network clusters by observation class, reflecting the inherited radial-distance encoding. (b) Training produces finer within-class subclusters; probe results (Table 9) show these encode visit history rather than node identity. Compare with the binary tree (Figure 3), where within-class subclusters encode spatial position.

individual nodes. t-SNE embeddings (Figure 7) confirm that training fragments observation-class clusters into subclusters encoding visit-history contexts rather than spatial position.

Table 9: Radial arm maze: inherited vs. learned structure (5 seeds). The inherited component (PC1–radial $\rho$) replicates the binary tree pattern. The learned component is visit history, not spatial discrimination: the node probe is flat while visit probes show large trained gains.

| Metric | Raw Obs | Untrained | Trained |
|---|---|---|---|
| \|PC1–radial\| $\rho$ | — | $\sim 0.82$ | $0.862 \pm 0.002$ |
| Node probe (25-way) | 27.6% | 27.2% | $27.4 \pm 0.2\%$ |
| Visit-bit probe (per-arm) | 52.8% | $61.8 \pm 0.4\%$ | $72.9 \pm 1.1\%$ |
| $N$-visited probe (9-way) | — | $24.5 \pm 0.4\%$ | $52.9 \pm 6.5\%$ |
| Arm identity (8-way) | 12.8% | 12.4% | 12.9% |

The encoding swap replicates on this non-tree topology: under random encoding, the inherited radial-distance axis disappears while the node probe jumps to 76.1% (Table 10), confirming that the encoding-determined partition generalizes across graph structures.

PCA ablation produces an analogous dissociation: PC1 ablation disrupts radial position awareness, spiking the revisit rate (the analog of leaf amnesia in the tree), while PCs 2–4 ablation disrupts hub-level foraging decisions without causing revisiting errors.

## 4 Discussion

### 4.1 Inherited vs. Learned Structure

Structure in trained hidden states should not be attributed to learning without comparison against untrained baselines. In both environments studied here, the most prominent feature would have been misattributed

Table 10: Radial arm maze encoding swap (5 seeds). The inherited radial-distance axis toggles on/off with encoding type; node probe inverts, paralleling the binary tree pattern.

| Encoding | Condition | \|PC1–radial\| $\rho$ | Node Probe | Visit-Bit |
|---|---|---|---|---|
| Structural | Untrained | $\sim 0.82$ | 27.2% | $61.8 \pm 0.4\%$ |
|  | Trained | $0.862 \pm 0.002$ | $27.4 \pm 0.2\%$ | $72.9 \pm 1.1\%$ |
| Random | Untrained | $0.353 \pm 0.030$ | $56.6 \pm 0.5\%$ | $64.5 \pm 0.2\%$ |
|  | Trained | $0.474 \pm 0.029$ | $76.1 \pm 0.2\%$ | $69.9 \pm 4.1\%$ |

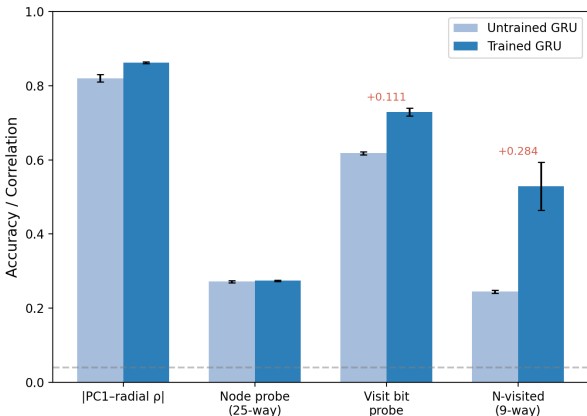

Figure 8: **Radial arm maze: inherited vs. learned structure.** The inherited component (PC1–radial $\rho$) is present before training. The learned component is visit history, not spatial discrimination: node probe and arm identity are flat, while visit-bit and $N$-visited probes show large trained gains. Compare with the binary tree (Table 5), where the node probe is the primary learned metric. Mean $\pm$ 1 std across 5 seeds.

without this comparison. The decomposition is consistent with an information-bottleneck interpretation: the cross-entropy objective pressures the network to retain policy-relevant information, and the input encoding already provides coarse geometric information while training refines task-relevant discrimination within aliased classes.

The encoding swap (Section 3.3) reveals a paradox: an inherited depth structure may limit how thoroughly the network builds within-class representations. When the encoding provides coarse spatial information for free, the behavioral objective is satisfied with less learned refinement; the network is under less pressure to develop its own spatial discrimination. This explains why removing the inherited scaffold produces dramatically richer learned representations while maintaining near-equivalent performance. The radial arm maze (Section 3.5) confirms this pattern generalizes to non-hierarchical topology, where the learned component is categorically different (visit history rather than spatial discrimination).

## 4.2 Illustrative Application: Grid Cell Emergence

As an illustrative application beyond discrete graphs, we apply the untrained baseline (Step 1 of our decomposition) to the architecture of Banino et al. (2018), the most widely cited result on grid cell emergence in trained networks. Dordek et al. (2016) proved that the non-negative principal components of difference-of-Gaussian (DoG) place cell inputs are hexagonal, and Sorscher et al. (2023) generalized the result via pattern formation theory and connected it to Banino-style architectures, raising the question of whether that structure already exists before training. The Banino architecture is a 128-unit LSTM with a 512-unit linear bottleneck projecting to 256 place cells and 12 head direction cells. We ran untrained forward passes (random weights, 10 seeds) in the original $2.2\,\mathrm{m} \times 2.2\,\mathrm{m}$ arena under three input encodings: Banino's velocity inputs, Sorscher softmax-Gaussian place cells, and Sorscher softmax-DoG place cells.

Table 11: Untrained-baseline gridness across input encodings (10 seeds, shared across encodings so the comparison is paired); scored with a port of `google-deepmind/grid-cells/scores.py`. Threshold-pass rate (% of units with gridness $\geq 0.37$ per Banino's published criterion) is reported alongside the raw mean gridness score; the 0.0% velocity entry is not a thresholding artifact, as the raw distribution has a negative mean and no unit-seed score approaches 0.37. Paired BCa 95% CI on the (DoG $-$ velocity) raw-mean difference: $[+0.48, +0.50]$; on the (Gaussian $-$ velocity) raw-mean difference: $[+0.42, +0.44]$. The Sorscher place cell encodings produce substantially higher gridness than velocity inputs, attributable to the encoding's spatial autocorrelation structure alone.

| Input encoding | $\% \geq 0.37$ | Raw mean gridness |
|---|---|---|
| Velocity (Banino's input) | $0.0 \pm 0.0$ | $-0.241 \pm 0.003$ |
| Sorscher softmax-Gaussian | $22.5 \pm 2.2$ | $+0.189 \pm 0.018$ |
| **Sorscher softmax-DoG** | **$26.9 \pm 1.7$** | **$+0.249 \pm 0.010$** |

The untrained baseline validates Banino's original result for their intended inputs. Under velocity, the untrained network produces 0.0% grid-like units while training produces $23\%\pm2.8\%$ (Banino et al., 2018), and the grid cells in that setup are genuinely learned. Under Sorscher's biologically motivated place cell encodings, however, the untrained network produces high gridness without any training (Table 11), $22.5 \pm 2.2\%$ for Gaussian and $26.9 \pm 1.7\%$ for DoG. Holding architecture and training regime constant (all untrained, all Banino-style LSTM), the encoding alone moves the untrained network from no gridness (0% under velocity) to substantial gridness (27% under DoG), a 27 percentage-point lift attributable to the input encoding's spatial autocorrelation structure.

The Dordek et al. (2016) theorem applies to the *non-negative principal components* of DoG place cell activity; we observe that the autocorrelation-driven component of this result survives random ReLU projections of the same input, even without the non-negative-PCA construction. This is consistent with Sorscher et al. (2023)'s observation that the grid-score distribution in Banino et al. (2018) was indistinguishable from that of low-pass-filtered random noise. The periodicity in the bottleneck need not reflect a learned hexagonal structure. Hexagonality in the bottleneck is also not the same as linearly accessible position. A two-layer MLP recovers the agent's position from the untrained bottleneck at $R^2 = 0.999$ (held-out, 80/20 timestep split) across all three encodings, but a ridge linear probe ($\alpha = 1$) achieves only $R^2 = 0.05$ on the DoG bottleneck (the encoding with the highest gridness), compared with $R^2 = 0.97$ for the Gaussian encoding. The linear-decoding gap partly reflects the nonlinear-by-construction nature of softmax DoG inputs. What stands out is that the network simultaneously carries highly hexagonal spatial firing structure and a position code requiring a nonlinearity to read out, in an untrained random projection.

This parallels our binary tree finding. Whether structure is inherited or learned depends on the input encoding, our central claim now generalized to continuous space and a different analysis metric. Under canonical scoring, the hexagonal grid structure in the Banino architecture under Sorscher place cell inputs arises from the input encoding alone, without training. An untrained baseline check is therefore informative whenever spatially structured observations are used as network inputs and gridness is measured in hidden layers, formalizing the concern raised by Sorscher et al. (2023) that the encoding can donate periodic structure before any learning occurs.

### 4.3 Implications for POMDP Methods

The architectural comparison (Section D) reveals that gated recurrence concentrates task-relevant features on a single principal axis while attention distributes equivalent information across dimensions, a representational structure trade-off rather than merely a capacity one.

The Bayesian filtering comparison places the trained GRU between the factorized memoryless-action filter and the joint-state filter with first-order action smoothing. The action-prediction probes are consistent with the GRU partially recovering action-history information that the factorized filter cannot represent.

This complements Ni et al. (2022)'s finding that tuned recurrent policies are competitive with specialized POMDP methods, and locates the source of that competitiveness in implicit action inference rather than a capacity advantage. The radial arm maze's 15× larger GRU–MLP gap further demonstrates that the value of recurrence depends heavily on task structure.

### 4.4 Limitations

Several limitations bound the scope of our conclusions. First, although the radial arm maze extends the framework beyond trees, both environments are small, discrete, and synthetic. Real animals navigate using rich multisensory cues, and both environments use hand-designed observation encodings. Second, the dataset is small (1,578 bouts from 10 mice), the leave-one-mouse-out analysis is underpowered ($n = 2$), and the water-unrestricted null result is based on a single cohort. Third, the training dynamics analysis uses a single optimizer and hyperparameter configuration; the compression-then-expansion trajectory may differ under other optimization choices.

## 5 Related Work

### 5.1 Emergent Spatial Representations in Neural Networks

Neural networks trained on spatial tasks spontaneously develop representations resembling hippocampal and entorhinal codes, including grid-like firing patterns (Banino et al., 2018; Cueva & Wei, 2018; Sorscher et al., 2023). These studies generally report trained representations without controlling for input encoding contributions; Section 4.2 applies our untrained baseline to the Banino et al. (2018) architecture, correctly validating their result while showing that other input encodings produce substantial untrained gridness that an analysis without an untrained-baseline check would attribute to training. The Tolman-Eichenbaum Machine (Whittington et al., 2020) addresses this by factorizing structural and sensory codes on arbitrary graphs. Benna & Fusi (2021) framed spatial representations as a byproduct of efficient memory compression, consistent with our finding that behavioral cloning yields depth encoding as a compressed sufficient statistic. More broadly, Lampinen et al. (2024) show that learned representations are systematically biased toward simpler features in complexity, learning order, and prevalence, with simpler features explaining more representation variance (including top principal components) regardless of the computational importance of harder ones. Our work complements this by intervening directly on the input encoding while holding architecture, training procedure, and trajectory data constant, and observing how the inherited/learned decomposition responds. The binary tree and grid cell case studies show the partition is encoding-determined: a structurally aliased low-dimensional encoding (5 unique observations across 127 nodes) inherits depth structure as strongly as a richer encoding does, and a random encoding of matched dimensionality and aliasing ratio destroys it. We offer the decomposition as a quantitative framework applicable beyond these two cases. On the biological side, hippocampal splitter cells (Wood et al., 2000; Frank et al., 2000) fire differently at the same location depending on trajectory history, predicting context-dependent representations in networks navigating aliased environments. Sun et al. (2025) recently showed that hippocampal activity during learning progressively orthogonalizes into state-machine-like representations.

### 5.2 Navigation Under Partial Observability

The radial arm maze (Olton & Samuelson, 1976) is a classic paradigm for spatial working memory requiring episodic visit tracking. Recurrent networks have long been applied to partially observable navigation (Chrisman, 1992; Bakker, 2001; Hausknecht & Stone, 2015), and Ni et al. (2022) showed that well-tuned GRU/LSTM policies are competitive with specialized POMDP methods. The Clone-Structured Cognitive Graph (CSCG; George et al., 2021) addresses aliasing through multiple "clones" per observation; Raju et al. (2024) and Dedieu et al. (2024) extended this to higher-order sequence learning and Transformer architectures, respectively. Our GRU achieves analogous context-dependent state splitting through implicit recurrent dynamics. Prior mechanistic RNN analyses inform our approach: Maheswaranathan et al. (2019) showed convergence to low-dimensional line attractor dynamics, and Karpathy et al. (2015) found individual LSTM cells tracking bracket depth.

### 5.3 Inverse Reinforcement Learning for Animal Behavior

Maximum Causal Entropy IRL (Ziebart et al., 2008; Ziebart, 2010) has been applied to the labyrinth of Rosenberg et al. (2021) to recover time-varying rewards (Ashwood et al., 2022), model discrete latent states (Jha et al., 2024), hierarchical intentions (Zhu et al., 2024), and history-dependent policies (Ke et al., 2025). All assume full state observability, sidestepping the representational challenge the animal faces. Our primary experiments bypass this by training a recurrent network from aliased observations; the T-maze experiment uses IRL-derived targets under genuine partial observability.

## 6 Conclusion

We develop and apply a three-step decomposition for representation analysis of trained sequential models: (1) compare against untrained baselines to isolate input-driven structure, (2) compare against information-theoretic bounds to quantify what is achievable without learning, and (3) use causal interventions to test whether the encoding is functionally used. Applied to GRUs trained on aliased navigation, this decomposition reveals that the most prominent hidden-state structure, a depth-dominated geometry on PC1, is largely inherited from the input encoding rather than learned. A controlled encoding swap confirms the partition is encoding-determined: random observation classes eliminate the inherited depth axis while the GRU compensates with substantially greater learned discrimination and near-equivalent performance. Applied to a radial arm maze, the framework recovers an analogous inherited axis but qualitatively different learned structure (visit history tracking rather than node discrimination), confirming generality across environments with different topology and task demands.

As an illustrative application beyond discrete graphs, the untrained baseline applied to the continuous-space architecture of Banino et al. (2018) with gridness scoring as the analysis metric reveals analogous encoding-driven structure (Section 4.2). When the encoder is learned jointly with the recurrent policy, the inherited/learned boundary shifts: the untrained baseline must encompass the untrained encoder, and the inherited component reflects random feature extraction rather than designed encoding structure. Whether the decomposition remains as cleanly informative in that regime is an open empirical question, and extending the framework to networks processing richer observation models (visual or proprioceptive input) is the most important direction for future work.

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

## A    Training Hyperparameters

Table 12 summarizes all training hyperparameters for both stages of the pipeline.

## B    Dimensionality Sweep

Table 13 reports full dimensionality sweep results. Figure 9 shows temporal formation and dimensionality scaling.

## C    Training Dynamics Details

To characterize how the spatial encoding develops, we checkpoint the GRU at 10 points during training (epochs 0, 1, 2, 5, 10, 20, 50, 100, 150, 200) and evaluate probe accuracy, PC1–depth correlation, and within-class distance ratio at each (Figure 10).

Table 12: Training hyperparameters for both pipeline stages.

| Parameter | IRL Stage (T-Maze) | GRU Stage |
|---|---|---|
| Optimizer | Adam | Adam |
| Learning rate | 0.01 | $3 \times 10^{-4}$ |
| LR schedule | — | Cosine annealing |
| Epochs | 300 | 200 |
| Batch size | Full dataset | 64 |
| $\gamma$ (discount) | 0.99 | — |
| Soft VI iterations | 200 | — |
| $L_2$ regularization ($\lambda$) | 0.01 | — |
| Reward clamping | $[-10, 10]$ | — |
| Gradient clipping | — | Norm 1.0 |
| Sequence chunk length | — | 200 steps |
| Hidden state detachment | — | At chunk boundaries |
| Random seeds | — | 5 |

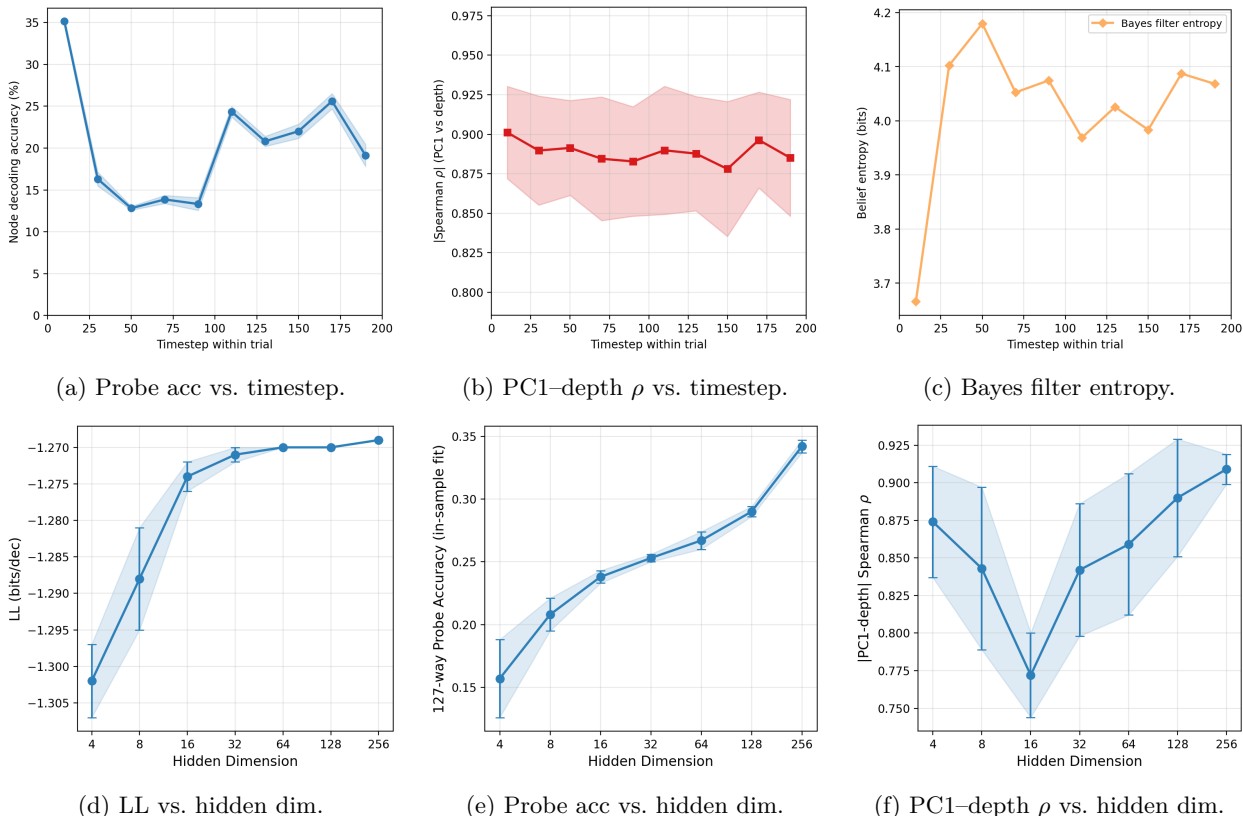

(a) Probe acc vs. timestep.  (b) PC1–depth $\rho$ vs. timestep.  (c) Bayes filter entropy.

(d) LL vs. hidden dim.  (e) Probe acc vs. hidden dim.  (f) PC1–depth $\rho$ vs. hidden dim.

Figure 9: **Temporal formation (top) and dimensionality sweep (bottom).** Top row: PC1–depth $\rho$ is flat from the first timestep, reflecting input structure; probe accuracy varies with accumulated trajectory history. Bottom row: log-likelihood saturates at $d \approx 32$, but probe accuracy continues to improve through $d = 256$, while PC1–depth $\rho$ remains high at all dimensionalities. Mean $\pm$ 1 std across 5 seeds.

Table 13: Dimensionality sweep results (mean $\pm$ std, 5 seeds). Log-likelihood saturates at $d \approx 32$; probe accuracy scales through $d = 256$. Probe accuracy here is reported as in-sample fit of the 127-way logistic probe on the hidden states it was trained on; the held-out probe accuracy at $d = 128$ (the main-paper configuration) is 25.7% (Table 5). The in-sample numbers are reported to characterize the dimensionality trend rather than as accuracy estimates.

| $d$ | LL (bits/dec) | Probe Acc | \|PC1–depth\| $\rho$ |
|---|---|---|---|
| 4 | $-1.302 \pm 0.005$ | $15.7\% \pm 3.1\%$ | $0.874 \pm 0.037$ |
| 8 | $-1.288 \pm 0.007$ | $20.8\% \pm 1.3\%$ | $0.843 \pm 0.054$ |
| 16 | $-1.274 \pm 0.002$ | $23.8\% \pm 0.5\%$ | $0.772 \pm 0.028$ |
| 32 | $-1.271 \pm 0.001$ | $25.3\% \pm 0.3\%$ | $0.842 \pm 0.044$ |
| 64 | $-1.270 \pm 0.000$ | $26.7\% \pm 0.7\%$ | $0.859 \pm 0.047$ |
| 128 | $-1.270 \pm 0.000$ | $29.0\% \pm 0.4\%$ | $0.890 \pm 0.039$ |
| 256 | $-1.269 \pm 0.000$ | $34.2\% \pm 0.5\%$ | $0.909 \pm 0.010$ |

Probe accuracy rises gradually from $15.8\% \pm 0.5\%$ (untrained) to $25.7\% \pm 0.7\%$ at epoch 200, with no abrupt phase transition (Figure 10a). Nearly half of the total improvement occurs by epoch 10, after which gains decelerate. The per-class breakdown mirrors the converged pattern (Table 4), with the largest gains concentrating at depths 2–4 where behavioral diversity along the goal path is greatest. Log-likelihood converges by approximately epoch 50 while probe accuracy continues to improve through epoch 200, consistent with the dimensionality sweep observation that probe-decodable spatial structure scales beyond what the behavioral objective requires (Section 3.2).

PC1–depth $\rho$ follows a non-monotonic trajectory: it peaks early (epoch 5) as training sharpens the inherited depth axis, then dips as the network develops orthogonal within-class encoding, before partially recovering by epoch 200 (Figure 10b). The within-class to between-class distance ratio reveals a complementary compression-then-expansion dynamic (Figure 10c): the ratio drops to a minimum at epoch 10 as depth structure sharpens, then rises steadily as nodes within the same observation class are pushed apart in hidden-state space. This provides a mechanistic account of how spatial discrimination develops incrementally from the cross-entropy signal, consistent with the depth-regression analysis (Section 3.2) showing that the learned encoding is largely orthogonal to the depth axis.

## D  Architecture Comparison Details

We train four recurrent architectures on the same data: GRU (Cho et al., 2014), LSTM (Hochreiter & Schmidhuber, 1997), Gated Transformer-XL (GTrXL; Parisotto et al., 2020), and minGRU (Feng et al., 2024). All use hidden dimension 64 and are trained for 200 epochs with 5 seeds (Table 14).

Table 14: Architecture comparison (hidden dim = 64, 5 seeds). All gated architectures match on behavioral fit; representational strategies differ. GRU $\approx$ LSTM on all metrics.

| Architecture | LL (bits/dec) | Probe Acc | \|PC1–depth\| $\rho$ | Params |
|---|---|---|---|---|
| GRU | $-1.270 \pm .000$ | $26.7\% \pm 0.7\%$ | $0.86 \pm 0.05$ | 30K |
| LSTM | $-1.270 \pm .000$ | $27.9\% \pm 0.4\%$ | $0.86 \pm 0.02$ | 38K |
| GTrXL | $-1.270 \pm .000$ | $26.8\% \pm 0.3\%$ | $0.46 \pm 0.12$ | 172K |
| minGRU | $-1.286 \pm .002$ | $21.7\% \pm 1.2\%$ | $0.94 \pm 0.02$ | 13K |

GRU and LSTM are nearly identical on all metrics: behavioral fit ($-1.270$ bits/dec), probe accuracy (26.7% vs. 27.9%), and PC1–depth $\rho$ (0.86 for both). The GTrXL achieves equivalent behavioral performance despite distributing depth information across dimensions rather than concentrating it on PC1 ($\rho = 0.46 \pm 0.12$). This demonstrates that equivalent task performance can arise from qualitatively different representational

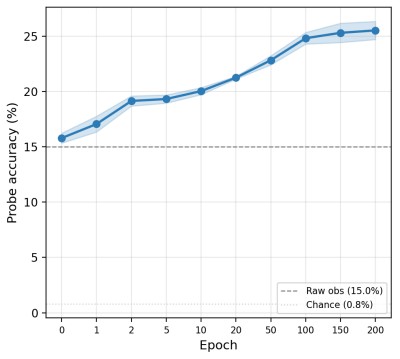 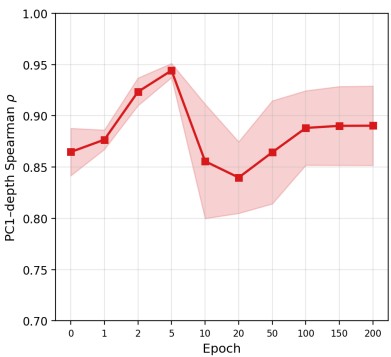 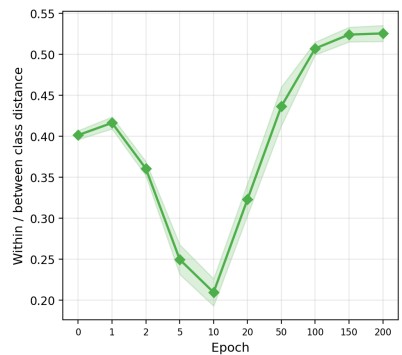

(a) Probe accuracy rises gradually with no phase transition; nearly half of the improvement occurs by epoch 10.

(b) PC1–depth $\rho$ peaks early then dips as the network develops orthogonal spatial encoding.

(c) Distance ratio shows compression then expansion: early depth learning followed by within-class differentiation.

Figure 10: **Training dynamics reveal gradual accumulation with compression-then-expansion.** (a) Node decoding accuracy rises monotonically with no phase transition. (b) PC1–depth correlation peaks early then dips as orthogonal spatial encoding develops. (c) Within/between-class distance ratio compresses then expands, reflecting a shift from depth learning to within-class differentiation. Mean $\pm$ 1 std across 5 seeds.

geometries, and that the concentrated depth-first abstraction is specific to gated recurrence rather than required by the task.

minGRU, with only 13K parameters, underperforms by $0.016\,\text{bits/dec}$ but achieves the highest PC1–depth concentration ($\rho = 0.94 \pm 0.02$). Its lower probe accuracy ($21.7\%$) suggests that extreme compression of depth onto a single axis may come at the cost of within-class spatial discrimination.

## E  T-Maze Control

A control experiment confirms that the depth-dominated geometry is not an architectural default. We apply the IRL-based variant (Section 2.2.2) to a passive T-maze, where the agent observes a cue at one end of a corridor and must turn left or right at a junction. The GRU achieves 100% junction accuracy, with PC1 encoding exactly the one bit the task demands (cue identity) rather than a depth-like gradient. This complements the encoding swap: the T-maze shows the geometry is task-specific, while the encoding swap shows it is encoding-specific.

## F  Boundary Conditions and Cross-Cohort Transfer

**Cross-cohort transfer.**  Leave-one-mouse-out cross-validation on two held-out animals (B1 and B2) yields negligible transfer gaps across log-likelihood, accuracy, and probe accuracy. With only $n = 2$ animals tested, this analysis is underpowered, but the qualitative finding is that the pipeline captures shared navigational structure rather than overfitting to individual behavioral strategies.

**Water-unrestricted mice.**  As a boundary condition, water-unrestricted mice from the same cohort, which show no water-seeking motivation and spend 65% of time at the root, yield no measurable advantage for recurrence over the MLP (both match the full-information ceiling at $-1.574\,\text{bits/dec}$). This suggests that structured, goal-directed behavior, not merely the presence of aliasing, is necessary for richer spatial encodings to emerge.

