# OpenReview forum: "Not All Structure Is Learned: Disentangling Inherited and Learned Representations in Recurrent Networks"
_TMLR — Accepted by TMLR_

### Review · Reviewer_FhgZ · 2026-04-22

**Summary Of Contributions:**

This paper studies in how far structure in trained recurrent representations is learned or partly inherited from the observation encoding, at the example of two navigation tasks from neuroscience. In a binary-tree maze and a radial-arms maze navigation task with strongly aliased state observations (inputs can be the same for different nodes of the maze), the authors compare trained and untrained Gated Recurrent Unit (GRU) networks, feed-forward memoryless baselines, Bayesian-filter baselines, and probe the hidden representation by linear decoders and PCA-based interventions. In the binary-tree task, the main empirical finding is that the dominant depth-like PC1 axis is already largely present before training, while training mainly adds within-class disambiguation. An experiment changing the input encoding, shows that similar performance can be reached but with a different, now learned representation of spatial locations. Similar results are reported for the radial-arm-maze extension.

Strengths

- The paper uses several complementary analyses rather than relying on a single visualization or metric.
- The encoding-swap experiment is a useful control, and the radial-arm-maze setting suggests that the results have at least some generality.
- Overall clear presentation of results.

Weaknesses

- That trained representations should be compared to untrained controls appears not surprising and not entirely novel, and it is claimed but not shown that omitting this is a wide-spread error in the literature. The core contributions are therefore not positioned clearly
- Some claims are broader or stronger than the evidence currently supports, especially language around information-theoretic bounds, double-dissociation, and the framework-like framing.

**Additional Comments:**

/

**Audience:**

Yes

**Audience Explanation:**

I think at least some readers working on recurrent models under partial observability of the environment, representation analysis, mechanistic interpretability, or computational-neuroscience-inspired analyses of learned state representations would be interested in the results.

**Broader Impact Concerns:**

/

**Claims And Evidence:**

No

**Claims Explanation:**

In the submitted version not fully, see requested changes.

**Requested Changes:**

**Major issues, most relevant for my recommendation:**

1) Clarify the Bayesian filter comparison.
Please state the exact assumptions and recursion for the “observation-only” filter, and explain in what sense it is Bayes-optimal. As written, it is difficult to interpret the claim that the GRU surpasses this filter on node decoding while also having access only to observations and not actions. If the GRU can infer past actions from observations, why can a observation-only Bayes-optimal method not? In how far are the observation-only, or the full-state Bayesian filter the technically correct information-theoretic limit to compare to?

2) Core contribution / novelty: What is the paper’s actual intended contribution beyond the generic point that trained representations should be compared to suitable controls? Please position this more carefully relative to prior work on representational bias and encoding-dependent geometry, especially Lampinen et al. and Sorscher et al.

3) Scope of the main claim: The paper currently suggests a broad, perhaps pervasive problem. Please either narrow the claim or provide concrete evidence that the issue materially changes interpretation in several existing recurrent-representation studies.

4) Ablation interpretation: Please sharpen the wording around “double dissociation.” In Figure 4, the most obvious separation appears at the leaves; elsewhere the two effects look similar, not distinct. In the text however, the interpretation evokes a general mid-depths vs leaves dissociation. Please provide further supporting analysis or make the claim more careful.

**Minor issues and optional suggestions:**

5) Action omission in the observation model: Why is previous action not included in the input encoding? Since part of the paper’s interpretation relies on inferred action history, this modeling choice seems important and should be justified, especially given the mouse-navigation setting: The mice would be expected to observe their last action.

6) In the introduction, the paragraph on memory+training does not clearly distinguish memoryless vs. recurrent (having memory also when untrained) from untrained vs. trained.

7) Please consistently define acronyms in the main text when used for the first time. Examples (non exhaustive): GRU, IRL

8) Why is the performance so similar across model comparisons? This seems to suggest that there is another principled bound which limits the performance of the models.

---

> ### Author Response · Authors · 2026-05-28
>
> Thank you for the thoughtful and thorough review. The table maps your points to the revision. Details follow.
>
> ### Summary
>
> | Point | Change | Where |
> |---|---|---|
> | #1 Bayes filter | Joint-state filter with action smoothing added, GRU reframed as intermediate | §2.3.4, §3.1, §4.3 |
> | #2 Positioning | Differentiation from Lampinen and Sorscher/Dordek sharpened | §5.1, §4.2 |
> | #3 Pervasiveness | "Pervasive" removed, framework reframed as diagnostic tool, §4.2 expanded | intro, §4.2, §5.1 |
> | #4 Double dissociation | Cross-seed paired analysis (5 seeds), CI excludes zero | abstract, intro, §3.4 |
> | #5 Action omission | Methodological note, action-prediction probe demonstrates recovery | §2.1, §3.1 |
> | Minor | Acronyms, three conditions distinguished, stochasticity floor noted | intro, §3.1, §3.2 |
>
> ### #1 Bayesian filter
>
> The factorized filter cannot represent action-history information the GRU may infer from observation context, so it was not the right bound. The revised paper adds a joint-state filter with first-order action smoothing, and the GRU now sits between the factorized filter and the joint-state ceiling rather than exceeding either (§3.1). The action-prediction probe is reframed as evidence the GRU partially recovers action history from observations. §4.3 locates the GRU's competitiveness with POMDP methods (Ni et al. 2022) in implicit action inference rather than capacity.
>
> ### #2 Positioning
>
> Lampinen et al. 2024 vary feature properties of the data and show simpler features dominate top PCs. We hold those properties implicit and intervene on the input encoding, with architecture, training, and trajectory data constant. The encoding-swap (§3.3) is the concrete instantiation. Two GRUs given different encodings produce qualitatively opposite inherited/learned patterns at near-equivalent behavior. Dordek et al. 2016 proved non-negative PCs of DoG place cell inputs are hexagonal, and Sorscher et al. 2023 generalized this. Our adjacent observation is that the autocorrelation-driven component survives random ReLU projections without the non-negative-PCA construction (§4.2). The contribution is the three-step decomposition, not the claim that encoding matters.
>
> ### #3 Scope
>
> "Pervasive" is removed. The framework is reframed as a diagnostic tool. §4.2 applies the diagnostic to the Banino architecture. Under Banino's velocity inputs the untrained network produces no gridness (so the published result is genuinely learned), but under Sorscher's biologically motivated encodings the same untrained architecture produces substantial gridness without any training. We also report a position-decoding caveat. A two-layer MLP recovers position from the untrained bottleneck cleanly, but a ridge linear probe does not on the DoG bottleneck, so hexagonality in the bottleneck does not by itself imply linearly accessible position. Scoring uses a verbatim port of the DeepMind canonical scorer.
>
> ### #4 Double dissociation
>
> Your critique drove us through two stages. We first re-ran at n=2000 on a single trained network. The reach-water rates under PC1 and PCs 2-4 ablation were not significantly different on that one network, and we initially read this as a magnitude indistinguishability and softened the claim. We then re-ran across all 5 seeds with the seed-paired bootstrap that should have been there originally. The magnitude direction is recovered. PC1 ablation impairs water-finding more than PCs 2-4 ablation (paired BCa CI [−6.8, −0.9]pp, excluding zero), though a single-seed test can flip direction, so the cross-seed paired test is the appropriate level of inference. The exploration-pattern dissociation is robust across all 5 seeds individually (§3.4, Table 8).
>
> ### #5 Action omission
>
> §2.1 notes that the agent's own actions are deliberately excluded from the input, so action history must be inferred from observation context. The action-prediction probe (§3.1) shows the GRU does recover action history from observations alone. The mouse can observe its last action. The model deliberately withholds this signal, and the probe measures how much the GRU recovers from context, which is the question of interest for an observation-only decomposition.
>
> ### Minor
>
> GRU and IRL spelled out at first use. Memoryless MLP, untrained GRU, and trained GRU are now treated as three separate conditions in the probe baselines and filter comparison tables. On performance similarity, the environment has high behavioral stochasticity, so roughly half the action distribution is irreducible noise. The GRU closes 27% of the gap from MLP to the full-information ceiling, and the remainder is the stochasticity floor rather than unused capacity.

---

### Review · Reviewer_uDmf · 2026-04-24

**Summary Of Contributions:**

This paper focuses on the distinction between inherited and learned representations in recurrent networks (notably GRUs). The task of navigating through a maze is studied and modelled as a tree traversal. The question is whether the behavior of the model is the result of the learning or is inherited from the input encodings. To answer this question, the representations between an untrained and trained GRU are compared in several ways (representation similarity, linear probing, causal PCA, Bayesian belief filter). It appears that the representations inherited from the encodings are already useful and notably ensure node class discrimination. Training further allows within-class discrimination with distinct clusters between samples. Moreover, both inherited and learned representations are responsible for the final model's behavior, as shown by the PCA causal analysis. These findings are confirmed on a non-hierarchical radial arm maze.

**Additional Comments:**

Disclaimer: I am not an expert in the paper's main area. As such, I may have misunderstood some claims regarding the method, and or miss some relevant related works.

**Audience:**

Yes

**Audience Explanation:**

The problem tackled is of great interest in improving our understanding of representation learning. Overall, the paper provides valuable insights into the importance of input encodings and the interplay between what is learned and what is encoded. This can have meaningful impact into architecture design and the emphasis put into data encoding (whether in structured task, in NLP with tokenization or in computer vision tasks).

**Claims And Evidence:**

Yes

**Claims Explanation:**

The experimental setup is sound with a focus on the complex and sequential task of navigating a maze: this requires a structured representation, and the importance of data encodings matters a lot. The experiments are extensive, with several metrics taken into account and proper analysis to avoid confounders. What might be lacking is more experiments on the impact of the architecture. Since I am not an expert in the paper's area, I would appreciate it if the authors could discuss this matter. Maybe the architecture considered is the standard choice for this task. Otherwise, it would be interesting to see how bigger models or different architectures impact the conclusion (which should be agnostic to the model considered since the purpose is to study the role of the input encoding).

**Requested Changes:**

I do not have specific changes about the current submission. I believe the method is well motivated with convincing experiments and valuable insights. One recommendation would be to add a visualization of the maze navigation and of the associated tree.

---

> ### Author Response · Authors · 2026-05-28
>
> Thank you for the careful review and the two suggestions.
>
> **Maze visualizations**: Two schematic figures have been added: the binary tree labyrinth in methods §2.1 and the 8-arm radial maze in methods §2.4.
>
> **Architecture comparison**. The four-way comparison across GRU, LSTM, GTrXL, and minGRU on the binary tree task is in Appendix D, referenced from results §3.4 and discussion §4.3. All four architectures are held constant on training data and observation encoding. GRU, LSTM, and GTrXL reach matched behavioral performance (−1.270 bits/dec); minGRU underperforms slightly (−1.286). Despite this near-equivalent behavioral fit, representational geometry diverges: gated recurrence (GRU, LSTM, minGRU) concentrates depth on a single principal component (PC1, ρ ≈ 0.86–0.94), while attention (GTrXL) distributes depth information across dimensions (ρ = 0.46). The depth-first geometry we report in the main text is therefore architecture-specific rather than task-required. Discussion §4.3 describes this as a representational-structure trade-off rather than merely a capacity one.
>
> We chose the GRU because it is the standard recurrent baseline in the POMDP benchmark literature (Ni et al. 2022, which motivates our information-theoretic comparisons) and because it produces the cleanest PC1 depth gradient, which makes the inherited/learned decomposition easiest to illustrate. The GTrXL result shows that the framework is not architecture-bound: the depth information is still there, it is just spread differently.

---

> > ### Comment · Reviewer_uDmf · 2026-06-14
> >
> > Thank you for the explanations. I appreciate the additional visualizations and better understand the choice behind the GRU presented in the paper. My initial assessment and confidence stand.

---

### Review · Reviewer_6g8v · 2026-05-15

**Summary Of Contributions:**

**Contributions**
1. **Main message: input encoding matters for representation analysis.** The paper argues that structure observed in trained recurrent networks is often inherited from the input encoding rather than learned from data, and that analyses which skip controls may conflate the two. The authors highlight this in a behavioral cloning experiment, in which most of the correlation with quantities of interest can already be found before any training.
2. **A quantitative framework.** The authors propose disentangling inherited from learned structure by combining (a) untrained baselines with the same architecture and inputs to isolate the encoding's contribution, (b) Bayesian belief filters to identify what is achievable without learning, and (c) causal PCA ablations to test whether identified components are actually used by the policy. None of these tools are individually new, but their combination to rule out the input structure makes sense.
3. **Application to behavioral cloning on mouse navigation.** The authors apply the framework to GRUs trained to imitate real mouse trajectories in a 127-node binary tree labyrinth, extend it to an 8-arm radial maze, and use these settings to showcase the importance of the input encoding in representational analysis.

**Strengths**
1. **The methodology is sound.** The untrained baseline establishes the input-driven floor, the Bayesian filters establish the without-learning ceiling, and the PCA ablation provides causal evidence. Controls like the random-direction ablation and the selectivity check on probes rule out the obvious objections.
2. **Results are clearly communicated**. The paper, while sometimes lengthy, is overall well written. The framework is clearly motivated, the results are well explained, and the contributions are nicely positioned in the broader literature.

**Weaknesses**
1. **Limited scope, partly because the headline finding is unsurprising.** The fact that structured inputs produce structured hidden states is not that surprising. This doesn't make the paper wrong as having a clean protocol that operationalizes a widely-suspected concern is useful, but the main example used (the tree) makes it more obvious than it may actually be.
2. **The grid cell application may deserved more space**. Grid cell emergence in trained networks (Banino et al., 2018) is one of the most influential and widely-cited examples of "learned" representations in the neuroscience-adjacent ML literature, and insights on this setup may be of wider interest.

**Audience:**

Yes

**Audience Explanation:**

This paper is of interest to the NeuroAI crowd, and in particular to the subset focused on understanding learned representations in biological and artificial neural networks. This community is part of TMLR's audience.

**Claims And Evidence:**

Yes

**Claims Explanation:**

The evidence for each claim is appropriate and well-controlled.

**Requested Changes:**

If possible, I would appreciate a more detailed discussion and experimentation around the grid cell example, potentially taking the role currently played by the T-maze. If not possible, please argue why.

---

> ### Author Response · Authors · 2026-05-28
> **Response to review**
>
> Thank you for the constructive review and the suggestion to expand the grid cell case study.
>
> §4.2 of the revised manuscript ("Illustrative Application: Grid Cell Emergence") has been substantially expanded in response, with the grid cell case now the central continuous-space demonstration of the framework, complementing the binary tree.
>
> The expanded §4.2 runs the Step 1 of our decomposition (untrained baseline) on the architecture of Banino et al. 2018, scored with a verbatim port of the DeepMind canonical scorer (`google-deepmind/grid-cells/scores.py`). The headline numbers are in the gridness table:
>
> - Velocity inputs (Banino's original setup): 0.0 ± 0.0% gridness untrained, vs Banino's published 23 ± 2.8% trained.
> - Sorscher softmax-Gaussian place cell inputs: 22.5 ± 2.2% gridness untrained.
> - Sorscher softmax-DoG place cell inputs: 26.9 ± 1.7% gridness untrained.
>
> Holding the architecture and training regime constant (all untrained, all Banino-style 128-unit LSTM with 512-unit linear bottleneck), switching the input modality from velocity to Sorscher place-cell encodings raises canonical gridness from 0% to 22–27% with no training at all. The architecture, scorer, and seeds are shared across the three conditions; only the input encoding varies.
>
> §4.2 also reports a position-decoding caveat (hexagonality in the untrained bottleneck does not by itself imply linearly accessible position) and positions the result against Sorscher et al. 2023's noise-comparison critique and Dordek et al. 2016's analytical theorem.
>
> Thank you again for the recommendation.

---

### Author Response · Authors · 2026-05-28
**Update**

We thank the reviewers for their careful engagement with the paper. We have uploaded our response and revised submission. The main changes include:

1. Bayesian filter comparison rebuilt: a joint-state filter with action smoothing replaces the factorized observation-only filter (methods §2.3.4, results §3.1 Table 3, discussion §4.3).

2. Double-dissociation analysis redone across all 5 GRU seeds with paired bootstrap CIs (results §3.4 Table 8).

3. Grid cell case study expanded with within-pipeline Banino comparison and the DeepMind canonical scorer (discussion §4.2 Table 11).

4. "Pervasive" removed from the introduction; framework repositioned as a diagnostic tool, with Lampinen and Sorscher / Dordek differentiation made explicit in Related Work.

5. Probe selectivity null corrected and inheritance-ratio CIs added (results §3.2 Table 5).

6. Maze schematics added (methods §2.1, §2.4); architecture comparison flagged more prominently from results §3.4; GRU/IRL spelled out at first use; action-omission note in §2.1; hippocampal optogenetic prediction removed from the conclusion.

---

### Decision · Action_Editor_4HWc · 2026-06-18

**Recommendation:** Accept as is

**Audience:**

Yes

**Audience Explanation:**

Yes, TMLR’s audience would be highly interested in the findings of this paper. The reviewer consensus highlights that this work is of great value to the NeuroAI community and researchers focused on mechanistic interpretability, representation learning, and neural networks operating under partial observability.

**Claims And Evidence:**

Yes

**Claims Explanation:**

This paper introduces a three-step diagnostic framework designed to separate representation structure in recurrent networks. The submission’s claims are supported by accurate, convincing, and clear empirical evidence, explicitly verified by the unanimous consensus of all three reviewers. The core finding, that a GRU's prominent hidden-state depth gradient is inherited rather than learned, is robustly backed by the fact that untrained baseline networks capture 97% of the trained representation's correlation. To isolate these input-driven structures, reviewers agreed that the three-step diagnostic protocol forms a methodologically sound and extensive approach by combining untrained baseline floors, information-theoretic benchmarks, and causal PCA ablations.The authors also proved the generalizability of their claims across contexts through robust control experiments.